# Unveiling complex patterns: An information-theoretic approach to high-order behaviors in microarray data

Antonio Lacalamita[1,2], Alfonso Monaco[1,2]*, Grazia Serino[3], Daniele Marinazzo[4], Nicola Amoroso[2,5], Loredana Bellantuono[2,6], Marianna La Rocca[1,2], Tommaso Maggipinto[1,2], Ester Pantaleo[1,2], Emanuele Piccinno[3], Viviana Scalavino[3], Sabina Tangaro[2,7], Gianluigi Giannelli[3], Sebastiano Stramaglia[1,2☉], Roberto Bellotti[1,2☉]

**1** Dipartimento Interateneo di Fisica M. Merlin, Università degli Studi di Bari Aldo Moro, Bari, Italy, **2** Istituto Nazionale di Fisica Nucleare (INFN), Sezione di Bari, Bari, Italy, **3** National Institute of Gastroenterology S. De Bellis, IRCCS Research Hospital, Castellana Grotte, BA, Italy, **4** Department of Data Analysis, Ghent University, Henri Dunantlaan, Ghent, Belgium, **5** Dipartimento di Farmacia—Scienze del Farmaco, Università degli Studi di Bari Aldo Moro, Bari, Italy, **6** Dipartimento di Biomedicina Traslazionale e Neuroscienze (DiBraiN), Università degli Studi di Bari Aldo Moro, Piazza G. Cesare, Bari, Italy, **7** Dipartimento di Scienze del Suolo, della Pianta e degli Alimenti, Università degli Studi di Bari Aldo Moro, Bari, Italy

☉ These authors contributed equally to this work.

* alfonso.monaco@ba.infn.it

## Abstract

The information-theoretic approach can shed light on the role of groups of correlated elements within a network. While there are already established methods for measuring new information, storage and transmission, the definition and application of methods for measuring information change remains an unresolved challenge. The change of information in a network is associated with redundancy and synergy between systems that share information about a target. Redundancy involves shared information about the target that can be retrieved using the individual source systems, while synergy involves information that can only be obtained by sharing the systems. A more refined approach, called partial information decomposition (PID), separates the unique, redundant and synergetic contributions of the shared information. However, these contributions cannot be directly derived from the classical measures of information theory. In this work, we apply PID approach to publicly available microarray gene expression data from 2 different experiments derived from patients affected by HCC and ASD. By comparing sample and gene synergy clusters with classical correlation clusters, we uncover higher order behaviours, such as differential genes and enriched functions closely linked to diseases phenotype, that emerge with this novel approach. These findings and further applications of this approach to gene expression data could shed light on the genetic aspects related to physiological aspects of complex diseases.

**Data availability statement:** All relevant data are within the manuscript and its Supporting Information files.

**Funding:** NA, LB, ST, and RB have obtained funding for this work under the National Recovery and Resilience Plan (NRRP), Mission 4 Component 2 Investment 1.4-Call for tender no. 3138 of 16 December 2021 of the Italian Ministry of University and Research funded by the European Union–NextGenerationEU (award number/project code: CN00000013), and Concession Decree No. 1031 of 17 February 2022 adopted by the Italian Ministry of University and Research (CUP: D93C22000430001), Project title: "National Centre for HPC, Big Data and Quantum Computing". AM, TM and RB have obtained funding for this work under the project "Genoma mEdiciNa pERsonalizzatA –GENERA", local project code T3-AN-04 – CUP H93C22000500001, financed under the Health Development and Cohesion Plan 2014-2020, Trajectory 3 "Regenerative, predictive and personalized medicine" - Action line 3.1 "Creation of a precision medicine program for the mapping of the human genome on a national scale", referred to in the Notice of the Ministry of Health published in the Official Journal no. 46 of 24 February 2021. AM, MLR, TM, EP and RB were supported by the Italian funding within the "Budget MIUR - Dipartimenti di Eccellenza 2023 - 2027" (Law 232, 11 December 2016) - Quantum Sensing and Modelling for One-Health (QuaSiModO), CUP:H97G23000100001. The funders had no role in study design, data collection and analysis, decision to publish, or preparation of the manuscript.

**Competing interests:** The authors have declared that no competing interests exist.

## Introduction

Through the information-theoretic approach, we can gain insights into the role of correlated element groups in a network. There are already established methods for measuring new information, storage and transmission, but the definition and implementation of methods for measuring information change remains an open challenge. The change of information in a network is related to redundancy and synergy between systems that provide information about a target. Redundancy means that each system independently provides the same information about the target, while synergy means that some information can only be obtained by combining the systems [1]. A classical method called interaction information decomposition (IID) analyzes information modification through the balance between redundancy and synergy [2–4]. However, the IID approach has the limitation that regards redundancy and synergy as mutually exclusive concepts, using a single value to quantify the modification [5,6].

A more refined approach called partial information decomposition (PID) [7] separates the unique, redundant, and synergistic contributions of shared information. However, PID requires new definitions for redundancy, synergy and unique information that cannot be directly derived from the classical measures of information theory.

Consequently, various researchers have proposed alternative measures to define the components of PID, resulting in a variety of definitions [7–11]. The lack of consensus on the desired properties of PID measures is the main reason for this proliferation.

Another challenge is the difficulty of reliably estimating the measures used in IID and PID decompositions. The estimation of probabilities using histogram-based methods is fraught with errors [12–14]. Although there are techniques that can improve the estimation of information storage and [15–17], their effectiveness in changing information has not yet been proven [18–21].

Several works used PID as a novel tool to investigate biological and physiological aspects [22–27]; for example, Wibral et al. [28] investigated the developmental course of information modification in a culture of neurons in vitro. Their findings showed that information modification increased with maturation but decreased as redundant information became dominant between neurons. This suggests that the neural system initially developed complex processing abilities, but ultimately exhibited highly similar information processing between neurons, possibly due to the absence of external inputs. In conclusion, they emphasised the significant potential of PID and information modification analysis for a better understanding of neural systems. Ince et al. [29] introduced a new estimation method that combined copula-based statistical theory with a closed-form solution for the entropy of Gaussian variables. This method creates a comprehensive, efficient, flexible, and robust multivariate statistical framework. It provides effect sizes on a consistent scale, handles discrete and continuous variables (both unidimensional and multidimensional), and allows direct comparisons of behavioral and brain response representations across different recording modalities. They demonstrated the effectiveness of this estimation

as a statistical test in neuroimaging, that accounts for both discrete stimulus categories and continuous stimulus features. Furthermore, Park et al. [30] analyzed the representational interactions between dynamic audio and visual speech signals and found that different brain regions exhibit different types of representational interactions. In particular, using a novel information-theoretic measure, they discovered that redundant encoding in the left posterior superior temporal gyrus/sulcus and synergistic encoding in the left motor cortex showed a hight correlation with speech comprehension performance.

Our work aims to apply partial information decomposition to biological data, in particular to gene expression from microarray experiments performed in our previous works [31,32] using complex networks and machine learning. Based on previous results, we used PID to investigate different gene communities in more detail and to explore the presence of crucial hidden information that is ignored by classical approaches. To our knowledge, our study is the first application of this novel tool on microarray data. We studied two different use cases: hepatocellular carcinoma (HCC) and autism spectrum disorder (ASD) to assess the the potential generality of this approach regardless of the chosen disease. In addition, we compared synergistic gene clusters obtained by PID with bivariate clusters achieved using mutual information. Finally, we applied biological tools such as differential gene expression (DGE) analysis and enrichment analysis to assess the differences between the found clusters. We would like to emphasize that these methods are not the final goal of this work, but are only used as comparative tools.

This paper is organized into five different sections: in the Methods Section, we give an overview of the analyzed data and the applied methodologies. In the Results Section we show the results obtained with the application of PID methods to two different use cases HCC and ASD. In the Discussion Section, we discuss our results and in the Conclusion Section summarize our findings.

## Materials and methods

As described in the introduction section, we analyzed microarray gene expressions from 2 different experiments studied in two previous works [31,32] by means of complex network and machine learning methods. The analysis implemented in this paper starts from the results of these two previous studies. Our pipeline is shown in Fig 1. Panel A refers to the previous analysis, where we selected the most biologically informative communities. Then, panel B describes the new analysis in which we investigated the existence of biological important sub-communities of genes, building the adjacency matrix through two different metrics: mutual information and synergy and by using Partitioning Around Medoids (PAM) as clustering algorithm.

### Data description

**Use case 1: HCC.** The microarray dataset GSE102079 and GSE54236 were downloaded from the GEO database (http://www.ncbi.nlm.nih.gov/geo/). The GSE102079 dataset [33,34] contains gene expression data extracted from liver tissue of 152 patients, specifically containing: 152 tumor and 91 adjacent liver tissues from HCC patients and 14 adjacent liver tissues obtained from patients with metastasis of colorectal cancer who had not received chemotherapy. The GSE54236 dataset [35–38] comprises gene expression data from 156 samples, including 78 hepatocellular carcinoma (HCC) tumor tissues and 78 corresponding adjacent non-tumor tissues, analyzed using the GPL6480 Agilent-014850 Whole Human Genome Microarray. This dataset served as an independent test set. Raw data were normalized using robust multiarray analysis (RMA) [39]; this method implements a background correction of the original data; then, a log2 transformation and finally a quantile normalization.

**Use case 2: ASD.** The dataset used is GSE28475 downloaded also from GEO; it consists of 104 samples divided in two classes: 33 ASD and 71 control samples [40,41]. We implemented a preprocessing procedure to minimize batch effects using the *ComBat* function from package *sva* version 3.50 [42]. Then, data were *log*2 transformed and quantile normalized through the *lumiN* function from package *lumi* version 2.54 [43].

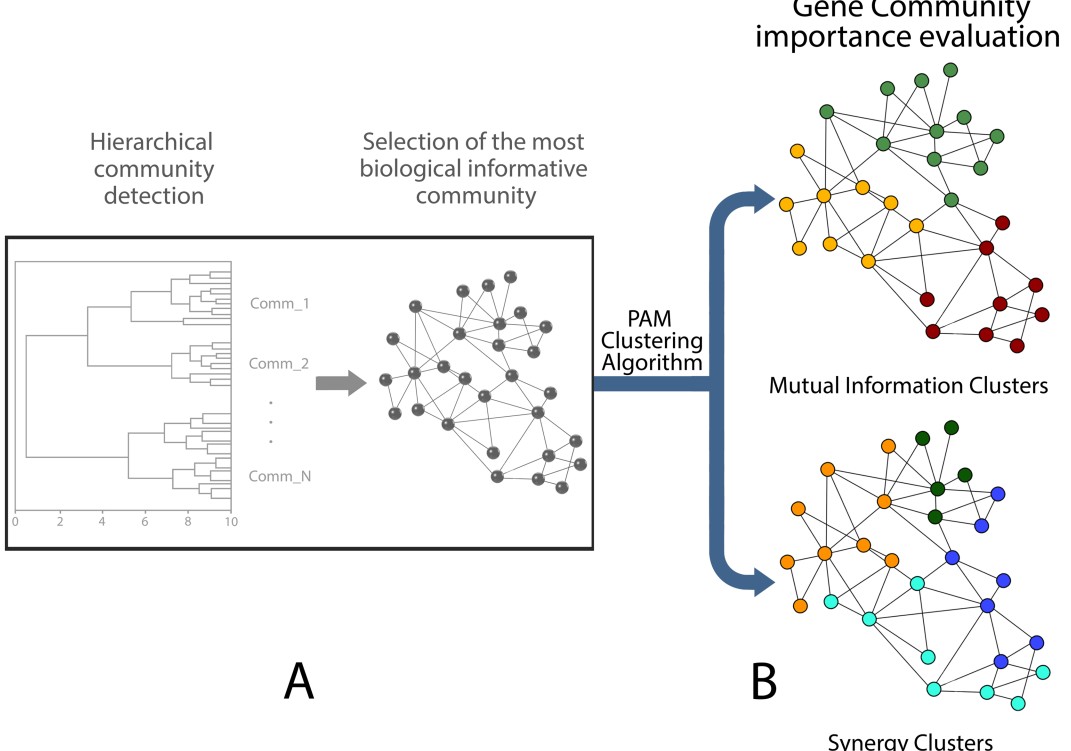

**Fig 1. Flowchart of the proposed analysis.** In panel A revisits our earlier analysis, where we identified the community with the most biological significance. Panel B, however, presents a new analysis in which we determined the biological importance of the gene community through two different grouping metrics based on mutual information and synergy. For this analysis, we employed the Partitioning Around Medoids (PAM) algorithm as our clustering technique.

## Mutual information

Entropy, the fundamental concept in information theory, characterizes the uncertainty or variability of a random variable [44–46] and thus is a useful measure of signal complexity [47]. Nevertheless, an engaging concept is the Mutual Information (MI). The MI between two variables quantifies their dependency and relationship using their entropy differences. Mutual information is a metric of the statistical dependency between two random variables, X and Y, and does not assume any particular distribution or type of interaction among the variables [48,49]. This freedom from restrictive assumptions makes MI very capable of finding nonlinear as well as nonmonotonic relationships. It can be represented using three mathematically equivalent formulations concerning entropy differences, each providing a different perspective on the information shared between the variables.

$$
\begin{aligned}
I(X;Y) &= H(Y) - H(Y|X) \\
&= H(X) - H(X|Y) \\
&= H(Y) + H(X) - H(X,Y) \\
&= H(X,Y) + H(X|Y) - H(Y|X)
\end{aligned}
\tag{1}
$$

Where $H(Y|X)$ represents the conditional entropy and $H(X,Y)$ is the entropy of the joint distribution of X and Y. MI can be interpreted as a standard statistical hypothesis test of independence, in many ways analogous to a t-test or correlation

test [50]. When regarded this way, absolute unbiasedness is less crucial; instead, emphasis lies on statistical significance evaluation and overcoming issues such as multiple comparisons [51]. MI is special in its advantages of sensitivity, robustness, and additivity. It does this by offering a general framework for analyzing discrete, continuous, and multi-dimensional variables using easily comparable measures of effect size on meaningful scales [52]. Despite all these beneficial properties, accurately estimating MI from bound experimental data is a quite difficult problem [53].

One of the ways to estimate mutual information is through the Gaussian Copula Mutual Information (GCMI) [29], which makes use of a copula that is a statistical construct that captures the dependency between two random variables independently of their marginal distributions [54] (https://github.com/robince/gcmi/tree/master/matlab).

## Partial information decomposition

In a situation where there are multiple source elements synapsing onto a single target element, the way information about the target is distributed amongst different combinations of source elements can be quite complex. To begin to examine this, consider a very simple "complex system" composed of two parent elements, $X_1$ and $X_2$, both synapsing onto a target $Y$. As already mentioned, mutual information $I(X_1, X_2; Y)$ captures the total information provided by both parents about Y. However, this overall measure does not explain how the information is divided among $X_1$ and $X_2$ individually.

In order to decompose the joint mutual information into its basic elements in order to get a detailed insight into how information is distributed. This will enable us to identify and explain all the ways in which information might be shared, uniquely contributed, or redundantly represented among the source elements. This concept forms the basis of Partial Information Decomposition (PID) [7].

Fig 2 shows how the different parts of partial information (redundant, unique, and synergistic) are represented in a Venn diagram in terms of joint and marginal mutual information elements for two source variables ($X_1$ and $X_2$) and a target variable ($Y$) [7].

A notable difference between the marginal mutual information and the unique information is highlighted: the marginal mutual information overlaps, with each counting the redundant (shared) information towards its own marginal mutual information.

$$I(X_1, X_2; Y) > I(X_1; Y) \cup I(X_2; Y) \tag{2}$$

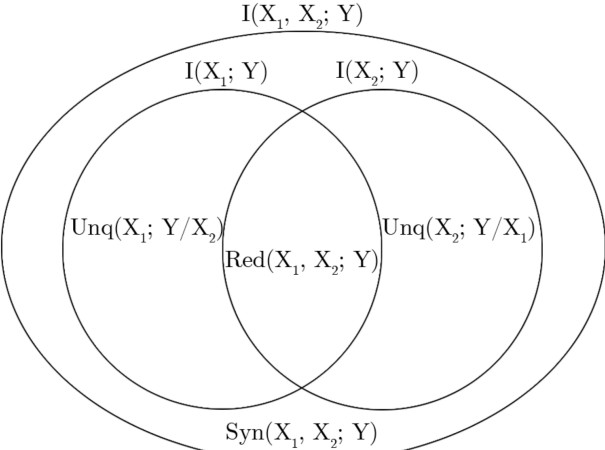

**Fig 2**. **The Venn Diagram comprises two circles, each representing the mutual information between one source variable and the target, while the large ellipse signifies the joint mutual information between both sources and the target.**

The discrepancy lies in the synergistic information, which cannot be attributed solely to either marginal mutual information.

The relationship between partial information terms and mutual information terms is delineated as follows:

$$Syn(X_1, X_2; Y) = I(X_1, X_2; Y) - I(X_1; Y) - I(X_2; Y) + Red(X_1, X_2; Y) \tag{3}$$

It's necessary to reintroduce a redundancy term because it is "double counted" when subtracting off the marginal mutual informations.

$$I(X_1, X_2; Y) = Red(X_1, X_2; Y) + Unq(X_1; Y/X_2) + Unq(X_2; Y/X_1) + Syn(X_1, X_2; Y)$$
$$I(X_1; Y) = Red(X_1, X_2; Y) + Unq(X_1; Y/X_2) \tag{4}$$
$$I(X_2; Y) = Red(X_1, X_2; Y) + Unq(X_2; Y/X_1)$$

The information can then be decomposed into specific components. $Red(X_1, X_2; Y)$ represents the information about Y that is *redundantly* available in both $X_1$ and $X_2$ (it can be observed through either $X_1$ or $X_2$). $Unq(X_1; Y/X_2)$ denotes the information about $Y$ that is uniquely provided by one of the sources, such as $X_1$, and is *only* observable through that specific source; similarly, the unique information $Unq(X_2; Y/X_1)$ is defined analogously. Finally $Syn(X_1, X_2; Y)$ is the information about $Y$ that emerges from the joint states of $X_1$ and $X_2$ and cannot be attributed to either source individually.

An issue with the Partial Information Decomposition (PID) lies in obtaining its constituent measures through classic information theory, necessitating an additional ingredient to provide an unambiguous definition of $Red(X_1, X_2; Y)$, $Unq(X_1; Y/X_2)$, $Unq(X_2; Y/X_1)$, $Syn(X_1, X_2; Y)$. Different redundancy and synergy definitions have been suggested to complete the PID definition [8,9,11]; the so-called minimum MI (MMI) PID is referenced [18], where Redundancy is defined as the minimum of the information provided by each individual source to the target. This choice ensures that Redundancy remains independent of the correlation between the source processes. Furthermore, under a joint Gaussian distribution of observed processes, all previously proposed PID formulations reduce to the MMI PID.

$$Red(X_1, X_2; Y) = \min I(X_1; Y), I(X_2; Y) \tag{5}$$

## Partitioning around medoids

The algorithm is designed to identify a series of points, called medoids, located in the centre of clusters [55]. These objects, called medoids, are summarised in a set $S$ of selected elements. Let $O$ represent the entire set of values, then $U = O - S$ is the set of unselected values. The goal of the algorithm is to minimise the average dissimilarity between each point and the nearest selected point. In other words, it aims to minimise the total dissimilarity between the objects and the nearest selected medoid. Partitioning Around Medoids (PAM) uses a greedy search strategy, which is faster than an exhaustive search, even if it does not always find the optimal solution. The algorithm consists of two steps: the BUILD step, in which it selects $k$ medoids from the $n$ data points in a greedy manner to minimise the cost that each data point associates with its nearest medoid; the SWAP step, in which the potential benefit in terms of cost changes due to the swap between a medoid and a non-medoid points is evaluated.

## Experimental procedure

Our experimental approach is based on the findings of two previous works [31,32]. In the two previous analyses we analyzed two different diseases, Hepatocellular Carcinoma (HCC) and Autism Spectrum Disorder (ASD), with a common pipeline based on complex networks and machine learning techniques. Specifically through a hierarchical community

detection approach based on the Leiden algorithm [56] we identified stable communities within the dataset. By applying a machine learning approach combining Boruta [57], Random Forest [58] and a 5-fold Cross Validation, we identified the communities that best discriminated healthy from diseased subjects. Then through eXplainable Artificial Intelligence techniques and functional enrichment we biologically validated these communities.

In particular in this work, we decided to further investigate two communities found in each previous analysis. We chose the community most interesting for biological aspect and the community without biological meaning but with the highest classification performance to discriminate sick from healthy patients. After selecting the gene communities, we applied the partial information decomposition explained in Sect , in two different configurations: in the first case we considered the subjects as nodes and in the second the genes as nodes of a network. If you take a closer look at the Venn diagram in Fig 2, the nodes become the subjects of our measures of mutual information ($X_1$, $X_2$, $Y$). To evaluate the higher order metrics, such as synergy, these measurements are made on each possible triple and the corresponding duets to fulfil the four equations in 4. Specifically, PID components (synergy, redundancy, and unique information) were calculated for each possible triplet combination of nodes (samples or genes) within a community, thus capturing higher-order interactions that traditional pairwise methods do not account for. In this way, for example, we obtained a three-dimensional matrix for synergy, which was subsequently averaged over the target size to obtain a dissimilarity matrix. We then calculated the silhouette to determine the optimal number of clusters, which we finally found using the PAM algorithm. The silhouette coefficient was evaluated for clustering solutions ranging from 2 to 10 clusters, selecting the number that maximized the silhouette score. We repeated the same procedure with the bivariate mutual information matrix and evaluated the differences inside the different groupings. For each subcommunity obtained by clustering, we assessed the accuracy for the independent dataset (where possible) and the enriched biological functions. Finally, to test the robustness of our analysis, we performed random sampling to understand whether accuracy and enrichment levels depend on Synergy or not. More specifically, we performed a bootstrap sample of genes for each considered community and then tested these random subcommunities as previously described to assess accuracy and statistically significant biological functions in independent test samples.

## Biological tool

To assess the biological significance of our results, we used complementary analyses: Differential Gene Expression (DGE) analysis and Pathways Analysis. For DGE analysis, we used the LIMMA R package [59], which applies linear modeling combined with empirical Bayes methods to detect statistically significant differences in gene expression levels between the identified clusters. Specifically, LIMMA was used exclusively to evaluate whether subsampling subjects through PID-derived groups could reveal additional differentially expressed genes compared to the entire original community. For Pathways Analysis, we performed Over-Representation Analysis (ORA) to identify functionally enriched biological pathways within predefined gene subsets identified through PID clustering. ORA is particularly suitable for assessing enrichment in gene subsets, as it does not require gene-ranking scores and evaluates the statistical significance of overlaps between gene clusters and predefined biological pathways. In detail, for the HCC use case, we conducted ORA using the Molecular Signature Database (MSigDB) via the GSEA software, considering hallmark gene sets, canonical pathways, gene ontology categories, and chemical and genetic perturbations [60]. Functions with a false discovery rate (FDR) below 0.05 were considered significantly enriched. For the ASD use case, we performed ORA using the Kyoto Encyclopedia of Genes and Genomes (KEGG) database [61–63] using the clusterProfiler package R [64]. Functions with a Bonferroni corrected p-value lower than 0.05 were considered significantly enriched. The biological validation of PID-derived clusters was achieved through this combined approach of identifying differential gene expressions and subsequently assessing pathway enrichments, explicitly demonstrating the biological significance of higher-order interactions identified through PID. The selection of specific enrichment tools for each use case (MSigDB using GSEA software for

HCC and KEGG via clusterProfiler for ASD) was consistent with the methodologies previously applied in our earlier studies [31,32], ensuring comparability between analyses.

## Results

### Use case 1: HCC

As mentioned above, our work starts from a previous analysis [31] in which we first implemented a community detection procedure using the Leiden algorithm to find stable gene communities within the gene co-expression network. Then, we applied an additional feature selection method based on the Boruta algorithm to each community found. In our present work, we focused on two different communities: community 29 the most biologically relevant and community 32 with highest HCC-controls classification accuracy on the test sample.

**Community 29.** Community 29 was composed of the gene expression of 51 genes belonging to 140 subjects. For this community, we performed two distinct analyses considering as nodes of a network first the samples and then the genes.

On the first analysis, with the 140 samples as the nodes, we implemented the Partial Information Decomposition based on all possible triplets, evaluating the higher order metrics. Then we compared the synergy and mutual information sample clusters obtained. In order to find the optimal number of clusters, we evaluated the silhouette metrics. Table 1 shows the label distribution within both synergy and mutual information clusters found by means of PAM algorithm.

Subsequently, to evaluate the biological meaning of our findings, we implemented a Differential Gene Expression (DGE) analysis. First of all we studied the Differential Genes (DGs) of the whole community and we considered a gene to be statistically differential expressed if the absolute log Fold Change value is greater than 1.5 and the adjusted P value is statistically significant at 1%. We adopted this logFC threshold in order to focus on genes with the largest and most biologically meaningful expression differences, providing a conservative filter that prioritizes robust disease-related signals over subtle but potentially less interpretable changes. Furthermore, we implemented this analysis to the clusters that combined both classes balance and cardinality. In Table 2 are displayed the number of DGs for all possible grouping indicating the number of information gained (+) or lost (-) compared to the whole community (the complete lists of genes are shown in the S1 Appendix of the Supporting Information).

As we can see, the synergy seems to maximize the variability of the samples: the SI clusters presents a mean number of DGs grater than the whole community and compared to two MI clusters. In particular in the SI clusters we found 2 genes linked to HCC biological functions: TCF21 [65] and RBP1 [66]. Both genes are connected with other genes notably deregulated in cancer as ANGPTL1, ADAMTSL2, PELI2 and EPCAM. To further validate these findings, we derived empirical p-values by comparing the observed gene-level statistics within synergy clusters against null distributions from

**Table 1**. Contingency matrix obtained from synergy (a) and MI (b) clusters of Community 29.

(a) Synergy contingency matrix.

|  | Cluster 1 | Cluster 2 |
|---|---|---|
| **Ctrl** | 14 | 43 |
| **HCC** | 47 | 36 |

(b) Mutual information contingency matrix.

|  | Cluster 1 | Cluster 2 | Cluster 3 |
|---|---|---|---|
| **Ctrl** | 31 | 3 | 23 |
| **HCC** | 14 | 34 | 35 |

**Table 2**. **Number of differentially expressed Genes for whole community 29 and for synergy and MI clusters.** In brackets we reported the number the number of DGs lost (-) or gained (+) compared to the whole community.

| Dataset | #DGs |
|---|---|
| Whole Community 29 | 7 |
| Synergy Cluster 1 | 8(+3, −2) |
| Synergy Cluster 2 | 7 |
| MI Cluster 1 | 8(+1) |
| MI Cluster 3 | 5(−2) |

1,000 matched random subject subsets (see Supplementary S1 Appendix). Among the candidate DEGs for both Synergy and MI, only TCF21 [65] remained significant under this more stringent test.

In the second part of our analysis, we considered the 51 genes as nodes of a network and we implemented the same pipeline in order to study the behaviors of the synergy gene communities with respect to the mutual information ones. The results of our analysis are shown in Table 3.

The lists of the new statistically significant enriched functions, gained over the entire community for MI and synergy clusters are shown in S1 and S2 Tables respectively of the Supporting Information. For the robustness test we bootstrapped the original community 10 times. No bootstrap resampling contains the enriched biological functions found with synergy. We reported these functions in S1 Appendix.

Finally, we built a classifier, based on Random Forest as in the original work [31], to test on an independent dataset the gene communities found with synergy and mutual information approaches. The classification performance among HCC and controls tissues are shown in Table 3: in term of accuracy the synergy cluster 2, shown in Fig 3, outperformed the two MI clusters found and the whole community with a reduction of approximately 60% of genes.

**Community 32.** Community 32 was composed of 41 genes belonging to 140 individuals. As for the previous community, we applied PID using both a complex network framework with subjects as nodes and a complex network framework with genes as nodes. The results of PAM clustering application are shown in Table 4.

In addition, we applied the DGE analysis to the three synergy clusters and to the first MI cluster, which appears to be much more balanced in terms of class than the second. Table 5 shows the number of DGs for the whole gene community and for each cluster found. Synergy clusters 2 and 3 have one DG (NAGS) more than the whole community like MI cluster

Table 3. **Machine learning classification performance for whole community 29 and for the found clusters on the independent dataset.**

|  | # genes | Accuracy | Sensitivity | Specificity |
|---|---|---|---|---|
| Whole Community 29 | 48 | 81.36 | 83.75 | 79.01 |
| Synergy Cluster 1 | 29 | 77.64 | 75.00 | 79.02 |
| Synergy Cluster 2 | 19 | 81.99 | 87.50 | 76.54 |
| MI Cluster 1 | 31 | 79.50 | 85.00 | 74.08 |
| MI Cluster 2 | 17 | 77.64 | 75.00 | 80.25 |

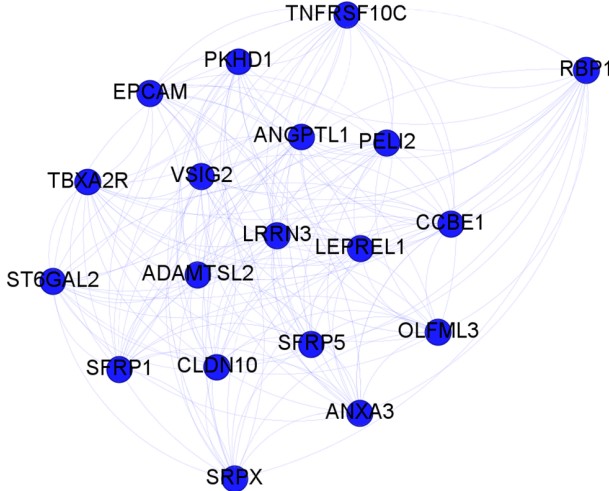

Fig 3. **HCC: Synergy gene cluster 2 from the community 29.**

**Table 4**. Contingency matrix obtained from Synergy (a) and MI (b) clusters of community 32.

**(a) Synergy contingency matrix.**

|  | Cluster 1 | Cluster 2 | Cluster 3 |
|---|---|---|---|
| **Ctrl** | 38 | 10 | 9 |
| **HCC** | 38 | 33 | 12 |

**(b) Mutual information contingency matrix.**

|  | Cluster 1 | Cluster 2 |
|---|---|---|
| **Ctrl** | 49 | 8 |
| **HCC** | 43 | 40 |

**Table 5**. **Number of differentially expressed genes for whole community 32 and for synergy and MI clusters.** In brackets we reported the number of DGs lost (-) or gained (+) compared to the whole community.

| Dataset | #DGs |
|---|---|
| Whole community 32 | 35 |
| Synergy Cluster 1 | 35 |
| Synergy Cluster 2 | 36(+1) |
| Synergy Cluster 3 | 36(+1) |
| MI Cluster 1 | 36(+1) |

1 (CYP7A1). These two genes appear not associated with the HCC phenotype and do not remained significant comparing them against null distributions (the complete lists of genes with their corresponding empirical p values are shown in the S1 Appendix of the Supporting Information).

Using genes as nodes in the complex network model and applying the PAM algorithm, we found 2 synergy clusters, shown in Fig 4, with a similar cardinality and 2 MI clusters much more unbalanced (see Table 6). As we can see in S3 Table of the Supporting Information, the SI clusters have new statistically significant functions associated with the HCC phenotype. Specifically, some of these biological functions were linked to hepatoblastoma and to the onset of liver cancer. We did not report any new novel biological processes for the MI clusters compared to the original community. As with community 29, we implemented a random forest model to assess the discriminatory power of these clusters in classifying HCC and healthy controls on the independent dataset. As shown in Table 6, a classification accuracy of 78.26% and

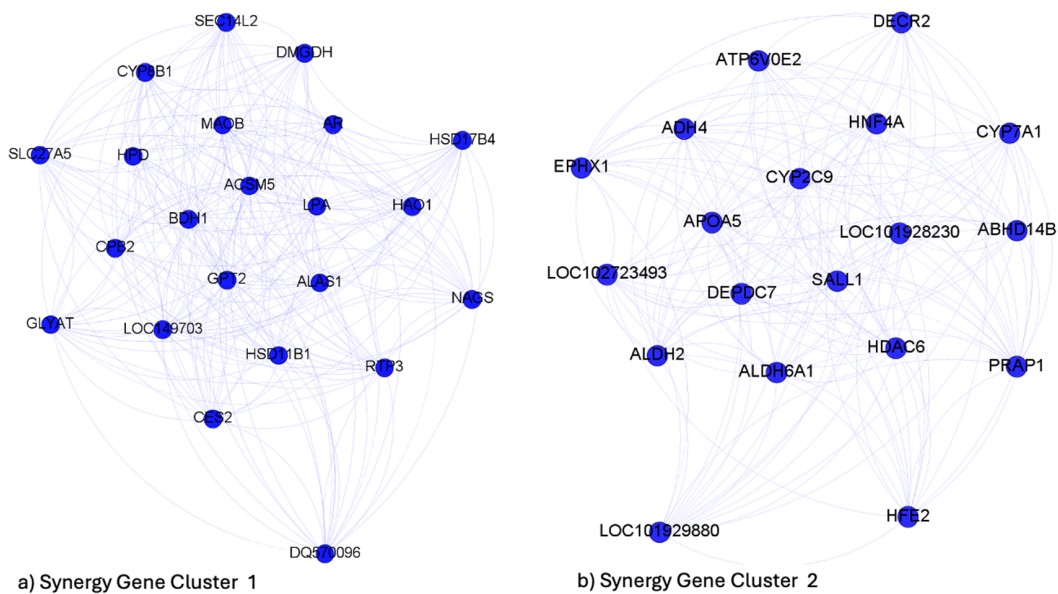

a) Synergy Gene Cluster 1  b) Synergy Gene Cluster 2

**Fig 4**. **HCC: Gene clustering based on synergy from community 32.**

**Table 6**. Machine learning performance for whole community 32 and for the found clusters on the independent dataset.

| | # genes | Accuracy | Sensitivity | Specificity |
|---|---|---|---|---|
| Whole Community 32 | 35 | 82.61 | 83.75 | 81.48 |
| Synergy Cluster 1 | 19 | 78.26 | 75.00 | 81.48 |
| Synergy Cluster 2 | 16 | 77.02 | 77.50 | 76.54 |
| MI Cluster 1 | 8 | 67.08 | 46.25 | 87.65 |
| MI Cluster 2 | 27 | 78.88 | 81.25 | 76.54 |

77.02% is achieved for the two synergy groups compared to 82.61% for the whole community; a slightly lower performance but with half the features used.

## Use case 2: ASD

As for the previous use case, we started from the results of a previous work where through a complex network and machine learning combined approach we selected some interesting gene communities respect to Autism phenotype. [32]. We decided to further investigate two communities: community 50 statistically and biologically relevant and community 78 that presented the highest performance in discriminating between autism and control subjects.

**Community 50.** Community 50 comprised gene expressions of 44 genes belonging to 104 subjects. As for the HCC use case we applied two different analyses in which we built two complex networks considering as nodes first the samples and then the genes. First, we developed a complex network framework with 104 nodes (the number of patients) and implemented Partial Information Decomposition based on all possible triplets, assessing the higher order metrics. Then we compared the synergy and mutual information of the pattern clusters. We choose the best number of clusters by evaluating the silhouette coefficient.

Table 7 shows the label distribution of both synergy and mutual information clusters found.

We implemented differential gene expression analysis for the biological evaluation of the detected clusters. For what concern the DGE analysis, no statistically significant differences are found with linear model from limma package [59], therefore we implemented a Kruskal-Wallis test [67] in order to compare the ASD gene expression distributions with the Control ones and to detect possible significant differences. We reported in Table 8 the number of genes with a statistically significant p-value at 1 % (the complete lists of genes are shown in the S2 Appendix of the Supporting Information).

In the synergy clusters we found 8 new DGs respect to the whole community. Instead 4 are the new DGs presented in the MI clusters:

**Table 7**. Contingency matrix obtained from synergy (a) and MI (b) of community 50.

**(a) Synergy contingency matrix.**

| | Cluster 1 | Cluster 2 |
|---|---|---|
| **Ctrl** | 33 | 38 |
| **ASD** | 26 | 7 |

**(b) Mutual information contingency matrix.**

| | Cluster 1 | Cluster 2 |
|---|---|---|
| **Ctrl** | 46 | 25 |
| **ASD** | 22 | 11 |

**Table 8**. Number of differentially expressed Genes for whole community 50 and for synergy and MI clusters. In brackets we reported the number of DGs lost (-) or gained (+) compared to the whole community.

| Dataset | #DGs |
|---|---|
| Whole community 50 | 6 |
| Synergy Cluster 1 | 8(+8, −6) |
| Synergy Cluster 2 | 3(−3) |
| MI Cluster 1 | 7(+2, −1) |
| MI Cluster 2 | 2(+2, −6) |

- Unique Synergy 8 DGs: SLC26A11, PPP1R9B [68], GPR150, PBX1 [69], RXRG [70], LOC286526, ZNF706, RUFY3;
- Unique MI 4 DGs: MGC40222, WAC [71], PRKCBP1, LOC286526.

Also in this case, to further validate these findings, we derived empirical p-values by comparing the observed gene-level statistics within synergy clusters against null distributions from 1,000 matched random subject subsets (see Supplementary S2 Appendix). Among the candidate DEGs, for both Synergy and MI, PPP1R9B [68], LOC286526, ZNF706 and FOXE3 remained significant under this more stringent test.

By using genes as nodes after applying Silhouette and PAM algorithm, we found 2 synergy clusters and 2 MI clusters as shown in Table 9. In synergy cluster 2, shown in Fig 5, we detected four different statistically significantly enriched functions as shown in S4 Table of the Supporting Information; no significantly enriched functions were found in MI clustering. For the robustness test we bootstrapped the original community 10 times. No bootstrap resampling contains enriched biological functions. After finding the biological content of synergy cluster 2, we could not test its classification performance in ASD control subjects because the overlap between the test sample and the cluster consisted of 3 genes only.

**Community 78.** In the network configuration with samples as nodes we found 2 synergy clusters and 2 MI clusters (see Table 10). The number of differentially expressed gene is summarized in Table 11. We excluded MI cluster 2 because

**Table 9**. Number of genes for whole community 50, synergy and MI clusters.

| Dataset | #Genes |
|---|---|
| Whole community 50 | 44 |
| Synergy Cluster 1 | 35 |
| Synergy Cluster 2 | 9 |
| MI Cluster 1 | 32 |
| MI Cluster 2 | 12 |

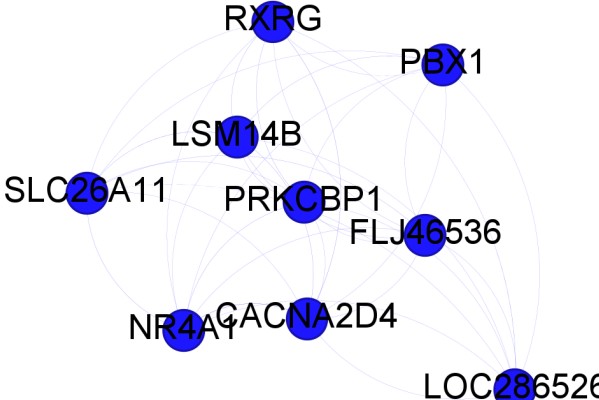

**Fig 5**. ASD: Synergy gene cluster 2 from community 50.

**Table 10**. Contingency matrix obtained from synergy (a) and MI (b) clusters of community 78.

| (a) Synergy contingency matrix. | | |
|---|---|---|
| | Cluster 1 | Cluster 2 |
| Ctrl | 54 | 17 |
| ASD | 14 | 19 |

| (b) Mutual information contingency matrix. | | |
|---|---|---|
| | Cluster 1 | Cluster 2 |
| Ctrl | 40 | 31 |
| ASD | 32 | 1 |

**Table 11. Number of differentially expressed genes for whole community 78 and for synergy and MI clusters.** In brackets we reported the number of DGs lost (-) or gained (+) compared to the whole community.

| Dataset | #DGs |
|---|---|
| Whole community 78 | 15 |
| Synergy Cluster 1 | 12(+1, −4) |
| Synergy Cluster 2 | 2(+1, −14) |
| MI Cluster 1 | 12(+4, −7) |

presented a very unbalanced configuration between ASD and control subjects (the complete lists of genes are shown in the S2 Appendix of the Supporting information).

The synergy and MI approaches highlighted 2 and 4 new DGs genes, compared to the whole community, respectively.

- Unique Synergy 2 DGs: CORO7, EVX2;
- Unique MI 4 DGs: OR52N2, MOGAT2, GP1BA, IL15RA.

As far as we know, these genes don't present a direct connection with ASD phenotype and only two genes (OR52N2 and MOGAT2) remain significant comparing them against null distributions (see S2 Appendix of the Supporting Information). By using genes as nodes after applying Silhouette and PAM algorithm, we found 4 synergy clusters and 2 MI clusters as shown in Table 12. In synergy cluster 3, shown in Fig 6, we detected five different statistically significantly enriched functions reported in S5 Table of the Supporting Information; we found nothing significant in the two MI clusters. For the robustness test we bootstrapped the original community 10 times. No bootstrap resampling contains enriched biological functions. After finding the biological content of synergy cluster 2, we could not test its classification performance in ASD control subjects because the overlap between the test sample and the cluster consisted of 4 genes only.

## Discussion

This work studied biological microarray data through partial information decomposition. We analyzed the information modification of two different use cases: hepatocellular carcinoma (HCC) and autism spectrum disorder (ASD). We started from the most interesting results of two our previous works to find gene communities with a biological meaning linked to the phenotype of two complex diseases. We used a clustering approach based on Partitioning Around Medoids (PAM) algorithm in which the adjacency matrix has been computed through two different metrics: mutual information and synergy. In both cases, we applied two different network configurations: samples as nodes and genes as nodes. Our results highlighted that, in a totally data driven process without any assumptions on the data composition and data characterization, grouping samples or genes detected considering a multivariate metric (synergy) respect to a bivariate one (Mutual Information) brought out additional biological patterns. We emphasize that PID is not intended as a feature selection method;

**Table 12. Number of Genes for whole community 78 and for synergy and MI clusters.**

| Dataset | #Genes |
|---|---|
| Whole community 78 | 57 |
| Synergy Cluster 1 | 5 |
| Synergy Cluster 2 | 29 |
| Synergy Cluster 3 | 17 |
| Synergy Cluster 4 | 6 |
| MI Cluster 1 | 45 |
| MI Cluster 2 | 12 |

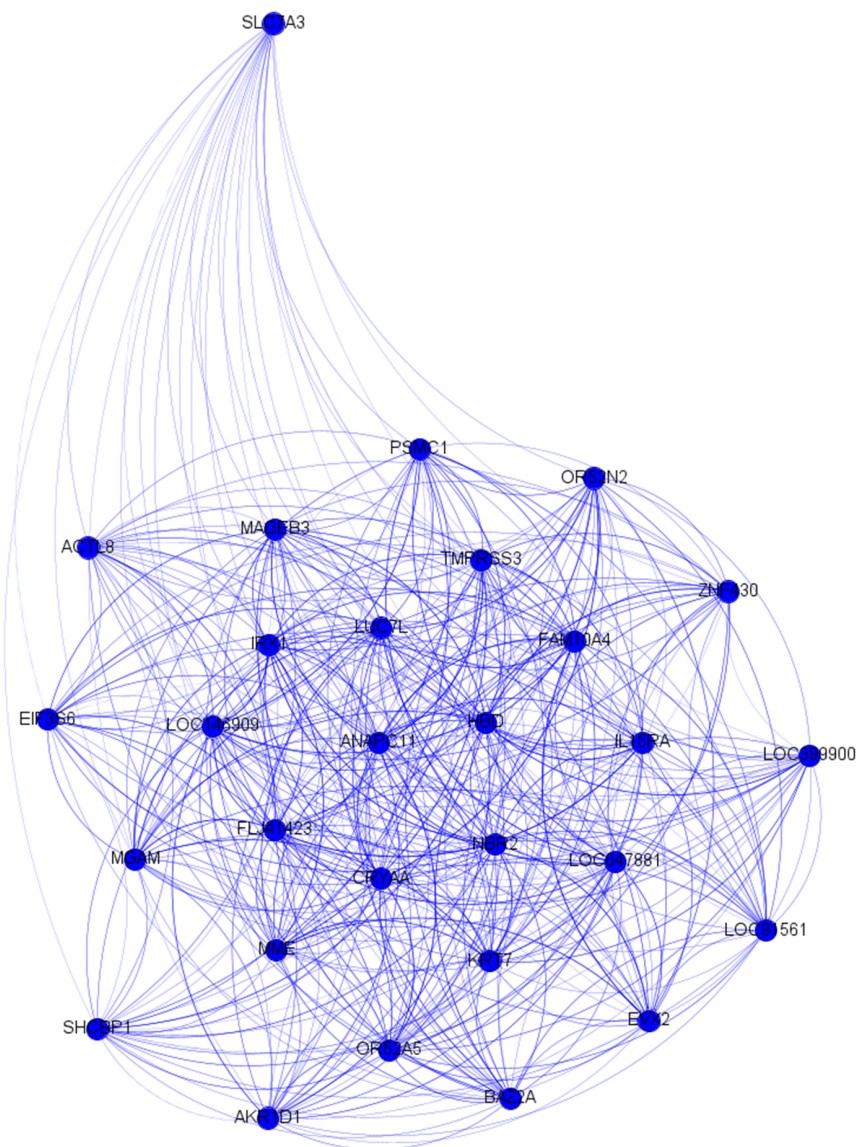

**Fig 6**. ASD: Synergy gene cluster 3 from community 78.

our focus was instead on revealing higher-order interactions that might uncover hidden biological insights. Classification performance was assessed only as a validation step to confirm the informational value of PID-derived substructures.

Analyzing the gene community 29 of HCC use case we found, among the differentially expressed genes of the synergy cluster, two genes connected to the tumour processes. TCF21, a member of the class II bHLH transcription factor super-family, has been shown to undergo abnormal methylation and is often inactivated in human cancers [65], and RBP1 that seems to be involved in different types of cancers including HCC [66]. These genes were also connected with other genes notably deregulated in cancer as ANGPTL1, ADAMTSL2, PELI2 and EPCAM. Moreover, enrichment analysis of genes in synergy clusters of community 29 indicated that these genes resulted silenced in adult cancers through DNA methylation mechanism. Several studies have reported that this mechanism contributed to the onset and progression of cancer. In fact, DNA methylation is a genetic modification that significantly influences cancer initiation and progression by silencing

tumor-suppressor genes through promoter hypermethylation and activating oncogenes through promoter hypomethylation [72,73]. Also in HCC, numerous studies have demonstrated the altered methylation status of genes in liver tissues of patients [74,75]. Furthermore, functions related to the *Wnt*/$\beta$-catenin signaling are enriched for the synergy cluster, this is one of the pathway frequently activated for HCC patients [76–78]; this pathway is also related to the differential gene RPB1 gained thanks to synergy grouping in the samples as nodes analysis [66].

Enrichment analysis of genes in the synergy clusters of community 32 revealed specific biological functions linked to the onset of pediatric and adult liver cancer. In fact, the first significant altered hallmark was hepatoblastoma [79] that was the most common pediatric liver cancer followed by other genes signature altered in hepatocellular carcinoma (HCC) such as SU_LIVER, LEE_LIVER_CANCER_E2F1_DN, LEE_LIVER_CANCER_MYC_E2F1_DN

[80,81], LEE_LIVER_CANCER_ACOX1_DN [82,83]. Notably, in our previous work, the enrichment analysis of the whole gene community detected no biological function associated to HCC despite its good statistical performance.

For both communities 32 and 29, no synergy and MI cluster outperforms the classification performance of the entire communities in the test dataset. Interestingly, synergy cluster 2 of community 32 matches the performance of the whole community for HCC-control subjects classification, but with a reduction of about 60% of the genes, emphasising how the synergy-based approach improves the discriminative power of the community by eliminating uninformative genes.

For ASD use case, in particular analyzing the synergy clusters of community 50, we detected as DG the gene RXRG linked to proper molecular patterns in the prefrontal and motor areas and also associated with the development of the prefrontal cortex-medial dorsal thalamus connection. RXRG is believed to be altered in ASD [69]. An other differentially expressed gene in synergy clusters is PBX1. In ASD probands, a de novo likely gene disruptive (LGD) variant and two de novo missense variants in the PBX1 gene were identified [84,85].

Additionally, two de novo loss-of-function variants and two rare, potentially damaging missense variants in the PBX1 gene were reported in ASD probands from the Autism Sequencing Consortium and the SPARK cohort [86,87].

As for the analysis with genes as nodes, for both original communities (namely communities 50 and 78) we found a synergy cluster enriched with a function statistically associated with the ASD phenotype. Conversely, all MI clusters found did not show any statistically significant biological function. For community 50, 4 new functions emerged from synergy cluster 2, which are listed in S4 Table. Two of these functions are related to the regulation of cortisol which is related to any kind of physical and mental stress. It is well the correlation with autism [88–93]. In particular, cortisol levels in the hypothalamic–pituitary–adrenal (HPA) axis were found to be higher in subjects with ASD than in healthy subjects [94,95]. The third function found, in order of statistical significance, concerns transcriptional dysregulation in cancer patients. Premature mortality in ASD seem due to various factors such as epilepsy, diabetes, cancer [96,97] and gastrointestinal diseases [98,99]. In particular, recent studies have found an overlap between biomarkers for ASD and various kind of cancers such as brain, thyroid and kidney [100–102]. The last significant function found concerns the mitogen-activated protein kinase (MAPK) signalling pathway, where deviations from the regular control were detected with several neurodevelopmental disorders including ASD [103–106].

For community 78, which had no biological significance, we discovered a synergy cluster (namely synergy cluster 3) with 5 enriched functions using the combination of PID and PAM, as listed in S5 Table. All these functions belong to a macrodomain related to gastrointestinal disorders (GID) [107–111]. Children with ASD very often develop critical conditions in the stomach that alter the intestinal epithelium and the composition of the gut microbiome (GM) [112–114]. There is growing evidence that the gut maintains bidirectional communication with the brain, creating the so-called microbiome-gut-brain axis, which is crucial for maintaining stable brain and gut function. Several studies have shown a link between GM problems and ASD; in particular, fats appear to play a special role in the diet of a child with ASD, as they can influence the GM community and the inflammatory response [115,116]. Another enriched function concerns the absorption of minerals: these are essential for the normal structure and key functions of the central nervous system and promote cell differentiation, development and migration [117,118]. Many studies showed links between vitamins, neurodevelopment and cognitive function by highlighting deficits in children with autism, [119–121]. In addition, several studies have shown

that differential metabolites in plasma can be used to distinguish cases of ASD from control subjects [122,123]: from these studies it emerged that a particular class of metabolites is enriched in studies of individuals with autism, namely glycerolipids [124]. Moreover, new evidence suggests that reduced mitophagy may improve Alzheimer's-related pathological features and behaviors. It is present both in postmortem brain samples from Alzheimer's patients and in animal models of the disease [125,126]. The involvement of neuronal mitophagy in the development of ASD has recently become a focal point of research [127,128]. Lastly, different studies have revealed that people with ASD who follow specific protein-restricted diets, such as wheat and dairy-free diets, have significantly lower intestinal permeability compared to those on unrestricted diets [129–134].

The information-theoretic and in particular partial information decomposition approach if applied to microarray data seems to be able to understand the mechanisms of complex diseases and to find potential targets for a future therapy. The superiority of PID in this study is highlighted by its unique ability to reveal internal structures within previously identified gene communities, which classical gene enrichment methods alone did not detect. Clearly, a more in-depth clinical validation is necessary to clarify how these genes are implicated in the biological processes linked to HCC and ASD.

However, our work has some limitations. First, PID is computationally complex since the measurements have to be performed for each triplet combination and the computation times are high, which limits the size of the network to be analyzed. Furthermore, this analysis can also be performed on higher orders, such as quadruplets, but this further increases the required computational power. Clearly, more in-depth clinical validation is needed to understand how these genes are involved in the biological processes associated with ASD and HCC, and application to other diseases may improve the validity of this analysis.

## Conclusion

In this article, we improved an information-theoretic method based on measuring the change of information in a network through synergy and redundancy to analyze gene communities associated with HCC and ASD. Based on the findings from 2 previous papers, we implemented in our article partial information decomposition (PID) and partitioning around medoids using microarray gene expression profiles from 2 different experiments. As far as we know, this is the first application of the PID method to microarray data. We found higher order behaviours, expressed as differential genes and enriched functions related to the progression of HCC and ASD. Our results demonstrate both the power of these techniques in the study of gene expression data and in the discovery of potential therapeutic targets for the treatment of complex diseases, although this also requires further investigation.

## Supporting information

**S1 Appendix. HCC: Additional information on the HCC analysis results.**
(PDF)

**S2 Appendix. ASD: Additional information on the ASD analysis results.**
(PDF)

**S1 Table. HCC: Comm 29 enrichment analysis on MI clusters.**
(PDF)

**S2 Table. HCC: Comm 29 enrichment analysis on synergy clusters.**
(PDF)

**S3 Table. HCC: Comm 32 enrichment analysis on synergy clusters.**
(PDF)

**S4 Table. ASD: Comm 50 enrichment analysis on synergy clusters.**
(PDF)

**S5 Table. ASD: Comm 78 enrichment analysis on synergy clusters.**
(PDF)

**S1 Fig. HCC: Comm 29 accuracy boxplot of boostrap samples.**
(PNG)

**S2 Fig. HCC: Comm 32 accuracy boxplot of boostrap samples.**
(PNG)

## Author contributions

**Conceptualization:** Alfonso Monaco, Daniele Marinazzo, Sebastiano Stramaglia.

**Data curation:** Antonio Lacalamita, Grazia Serino, Daniele Marinazzo.

**Formal analysis:** Antonio Lacalamita, Alfonso Monaco.

**Investigation:** Antonio Lacalamita, Daniele Marinazzo.

**Methodology:** Antonio Lacalamita, Alfonso Monaco, Daniele Marinazzo, Sebastiano Stramaglia.

**Software:** Antonio Lacalamita.

**Supervision:** Alfonso Monaco, Daniele Marinazzo, Gianluigi Giannelli, Sebastiano Stramaglia, Roberto Bellotti.

**Validation:** Antonio Lacalamita, Alfonso Monaco, Grazia Serino, Daniele Marinazzo, Sebastiano Stramaglia, Roberto Bellotti.

**Visualization:** Antonio Lacalamita, Alfonso Monaco.

**Writing – original draft:** Antonio Lacalamita, Alfonso Monaco, Grazia Serino.

**Writing – review & editing:** Antonio Lacalamita, Alfonso Monaco, Daniele Marinazzo, Nicola Amoroso, Loredana Bellantuono, Marianna La Rocca, Tommaso Maggipinto, Ester Pantaleo, Emanuele Piccinno, Viviana Scalavino, Sabina Tangaro, Gianluigi Giannelli, Sebastiano Stramaglia, Roberto Bellotti.

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
