## [Decision Letter · Decision Letter 0]

5 Jan 2025

PONE-D-24-38840Unveiling Complex Patterns: An Information-Theoretic Approach to High-Order Behaviors in Microarray DataPLOS ONE

Dear Dr. Monaco,

Thank you for submitting your manuscript to PLOS ONE. After careful consideration, we feel that it has merit but does not fully meet PLOS ONE’s publication criteria as it currently stands. Therefore, we invite you to submit a revised version of the manuscript that addresses the points raised during the review process.

The weaknesses of the manuscript include clarity of language, as some sentences are long and may cause ambiguity. For example, the phrase “To assess the biological significance of our results, we used two biological tools: Differential Gene Expression (DGE) analysis and Pathways Analysis” could be simplified for easier comprehension. The excessive technical language might make the manuscript less accessible to a broader audience. Regarding methodological innovation, while PID is an interesting approach, it is applied to previously published data, and the results do not seem to introduce major advancements in the field. The limitations of the PID method are not sufficiently discussed, especially in the context of extending its applications to other types of complex data. The analysis is restricted to only two clinical cases, and the lack of clinical validation or additional experiments reduces the practical applicability of the study. Additionally, the exploration of the data is sometimes superficial, and the biological implications of the findings are not adequately developed.

My recommendations include rephrasing certain paragraphs to improve clarity and avoiding excessive technical jargon. The authors could discuss the biological implications of their findings in greater detail and should address the limitations of the PID method more explicitly in the discussion section. The conclusions could be more clearly defined, emphasizing how these methods could be extended to other diseases or datasets. Adding further analyses on additional datasets would provide broader validity and relevance to the results. Interdisciplinary collaboration with clinicians or geneticists for the validation of the findings could add practical value and credibility to the study. 

We look forward to receiving your revised manuscript.

Kind regards,

Paul Aurelian Gagniuc, PhD

Academic Editor

PLOS ONE

Journal Requirements:

-A statistical framework for neuroimaging data analysis based on mutual information estimated via a gaussian copula (doi: https://doi.org/10.1002/hbm.23471)

-DNA methylation fingerprint for the diagnosis and monitoring of hepatocellular carcinoma from tissue and liquid biopsies (doi: https://doi.org/10.1101/2021.06.01.21258144)

(among others)

In your revision ensure you cite all your sources (including your own works), and quote or rephrase any duplicated text outside the methods section. Further consideration is dependent on these concerns being addressed.

4. Please ensure that you refer to Figure 5 in your text as, if accepted, production will need this reference to link the reader to the figure.

5. We note you have included a table to which you do not refer in the text of your manuscript. Please ensure that you refer to Table 1, 4 and 10 in your text; if accepted, production will need this reference to link the reader to the Table.

Reviewers' comments:

Reviewer's Responses to Questions

**Comments to the Author**

1. Is the manuscript technically sound, and do the data support the conclusions?

Reviewer #1: Partly

Reviewer #2: No

2. Has the statistical analysis been performed appropriately and rigorously? 

Reviewer #1: No

Reviewer #2: No

3. Have the authors made all data underlying the findings in their manuscript fully available?

Reviewer #1: Yes

Reviewer #2: Yes

4. Is the manuscript presented in an intelligible fashion and written in standard English?

Reviewer #1: Yes

Reviewer #2: Yes

5. Review Comments to the Author

Reviewer #1: General comment

In this paper, authors applied the partial information decomposition (PID) to two microarray gene expression datasets. In previous work, gene communities were identified. The PID is used to explore possible hidden information not accessible using classical methods. Results are shown for hepatocellular carcinoma (HCC) and autism spectrum disorder (ASD) microarray data.

Specific comments

In the material and methods section, authors should summarize their previous work results used in the present work. The content of the reference 32 may not be available to the reader and the Figure 1 is not enough to fully understand the previous work.

In the material and methods section, a Venn diagram is used to explain the partial information decomposition for two sources X1 & X2, and a target Y. However, there is a lack of link between the sources X1 & X2, the target Y, and the gene communities. It will be interesting to explain how the PID is applied to the gene communities to get the results presented.

How the redundancy and the synergy have been computed for the gene communities?

The X and Y in the mutual information are random variables which distributions are used to get I(X;Y). Are the sources X1 and X2 and the target Y random variables? How the mutual information is applied to clustering results and how the copula part is computed?

Authors should explain why it is necessary to perform clustering and why using the PAM clustering method and not another. Otherwise, that are the links between clustering results and the PID.

Reviewer #2: In this manuscript, the authors present an innovative approach of combining Partial Information Decomposition (PID) with microarray data to demonstrate that it can pinpoint mechanisms of complex disease. Their application, however, is limited to a set of genes that they have previously identified as capable of accurately distinguishing between disease and healthy states. This approach is highly limiting and is associated with two major problems.

1: The authors base their analysis on genes selected in their previous work (https://doi.org/10.3390/ijms242015286), making it challenging to critically assess the study without examining the gene selection process described in their prior research. According to their previous work, the gene communities identified are highly correlated. Highly correlated gene sets inflate redundancy. Thus, it is not surprising that subsets of redundant genes that are previously known to separate disease and healthy states would perform well. The claims that some of the synergy clusters outperform MI (Mutual Information) clusters and the whole gene set is not substantiated by evidence. Performance metrics summarized in Table 3 are very close to each other and without confidence intervals, it is not possible to evaluate the performance of SI (Synergistic Information) cluster genes.

2: The authors aim to demonstrate that when PID is applied to microarray data, it can pinpoint mechanisms of complex disease. Nonetheless, the data do not fully support this conclusion. For example, the authors claim that synergy maximizes the variability of the samples and leads to detection of a greater number of disease relevant genes. However, small differences in the number of differentially expressed genes derived from within cluster comparisons can be due to chance. The authors should combine bootstrapping and differential gene expression analysis to show that the small differences are not due to chance.

To avoid preselection biases, PID should be directly applied to all genes or to a biologically informed subsets (e.g., gene ontology).

Benchmarking MI against SI utilizing high-throughput data has potential; however, in it is current form, I cannot recommend this manuscript for publication.

Minor Issues:

- The Material and Methods section should include more detail including a codebase to encourage reproducibility. Also, all datasets used in the paper should be explicitly indicated in the Materials and Methods section. For example, GSE54236 is only mentioned in the Results section.

- The same statistical test for identifying differentially expressed genes should be used throughout the paper.

- “Transmission” is misspelled in the abstract.

6. PLOS authors have the option to publish the peer review history of their article (what does this mean?). If published, this will include your full peer review and any attached files.

Reviewer #1: **Yes: **Doulaye DEMBELE

Reviewer #2: No

---

## [Author Response · Author response to Decision Letter 1]

21 Feb 2025

I attached our response to reviewer and editor comments in the file point_to_point_answers_to_reviewers.pdf

---

## [Decision Letter · Decision Letter 1]

13 Jul 2025

PONE-D-24-38840R1Unveiling Complex Patterns: An Information-Theoretic Approach to High-Order Behaviors in Microarray DataPLOS ONE

Dear Dr. Monaco,

Thank you for submitting your manuscript to PLOS ONE. After careful consideration, we feel that it has merit but does not fully meet PLOS ONE’s publication criteria as it currently stands. Therefore, we invite you to submit a revised version of the manuscript that addresses the points raised during the review process.

We look forward to receiving your revised manuscript.

Kind regards,

Y-h. Taguchi, Dr. Sci.

Academic Editor

PLOS ONE

Journal Requirements:

Additional Editor Comments:

Since one of reviewers was very nagative, please try to address all the concerns raised by reviewers as much as possible.

Reviewers' comments:

Reviewer's Responses to Questions

**Comments to the Author**

1. If the authors have adequately addressed your comments raised in a previous round of review and you feel that this manuscript is now acceptable for publication, you may indicate that here to bypass the “Comments to the Author” section, enter your conflict of interest statement in the “Confidential to Editor” section, and submit your "Accept" recommendation.

Reviewer #1: (No Response)

Reviewer #2: (No Response)

2. Is the manuscript technically sound, and do the data support the conclusions?

Reviewer #1: Partly

Reviewer #2: No

3. Has the statistical analysis been performed appropriately and rigorously? 

Reviewer #1: No

Reviewer #2: No

4. Have the authors made all data underlying the findings in their manuscript fully available?

Reviewer #1: Yes

Reviewer #2: Yes

5. Is the manuscript presented in an intelligible fashion and written in standard English?

Reviewer #1: Yes

Reviewer #2: Yes

6. Review Comments to the Author

Reviewer #1: The contribution of this paper is the use of the PID to microarray gene expression analysis. However, the superiority of the PID approach is not shown over the methods used for gene set enrichment analysis. The calculation of the PID components from a gene community is not explained in the text and there is no biological validation of the findings.

The silhouette measure allows to evaluate a clustering method result. Authors should explain how they use the silhouette for determining the optimal number of clusters.

The differential gene expression analysis is usually done at the genome level to select a subset of genes for functional analysis. LIMMA is not a biological function enrichment tool.

Reviewer #2: The authors identified informative sets of subsamples by applying partial information decomposition (PID) on gene communities that was previously shown to distinguish healthy and disease states. Here, they also showed that when the informative cluster space is repeatedly bootstrapped using the same number of genes as the synergy cluster, the predictive performance of these random samples is comparable to that of the synergy cluster. This suggests that a random subsample of a similarly sized gene space can perform equally well in classifying healthy and disease states. Thus, PID does not present an advantage here as a feature selection method. These results should be clearly articulated in the results section and conclusions should be drawn aptly.

The authors also state that the synergy cluster is enriched for disease-relevant pathways based on GSEA. GSEA typically involves ranking all genes by the magnitude of differential expression whereas the authors appear to be comparing predefined gene subsets identified via PID. This suggests that the approach the authors might be using is over-representation analysis (ORA) for identifying functional enrichment. The enrichment method used should be clearly defined.

Additionally, I couldn't access supplementary Table 3. If authors had used ORA for identifying functionally enriched pathways in their subsets, it would be helpful to indicate the degree of overlap between the identified clusters and gene signatures. Same should be done for the boot-strapped samples.

Overall, the results don't support the strengths of PID.

7. PLOS authors have the option to publish the peer review history of their article (what does this mean?). If published, this will include your full peer review and any attached files.

Reviewer #1: No

Reviewer #2: No

---

## [Author Response · Author response to Decision Letter 2]

25 Jul 2025

We reported point to point answers to reviewers in the attached file point_to_point_answers_to_reviewers_new.pdf

---

## [Decision Letter · Decision Letter 2]

12 Aug 2025

PONE-D-24-38840R2Unveiling Complex Patterns: An Information-Theoretic Approach to High-Order Behaviors in Microarray DataPLOS ONE

Dear Dr. Monaco,

Thank you for submitting your manuscript to PLOS ONE. After careful consideration, we feel that it has merit but does not fully meet PLOS ONE’s publication criteria as it currently stands. Therefore, we invite you to submit a revised version of the manuscript that addresses the points raised during the review process.

We look forward to receiving your revised manuscript.

Kind regards,

Y-h. Taguchi, Dr. Sci.

Academic Editor

PLOS ONE

Journal Requirements:

Reviewers' comments:

Reviewer's Responses to Questions

**Comments to the Author**

1. If the authors have adequately addressed your comments raised in a previous round of review and you feel that this manuscript is now acceptable for publication, you may indicate that here to bypass the “Comments to the Author” section, enter your conflict of interest statement in the “Confidential to Editor” section, and submit your "Accept" recommendation.

Reviewer #1: All comments have been addressed

Reviewer #2: (No Response)

2. Is the manuscript technically sound, and do the data support the conclusions?

Reviewer #1: Partly

Reviewer #2: No

3. Has the statistical analysis been performed appropriately and rigorously? 

Reviewer #1: Yes

Reviewer #2: No

4. Have the authors made all data underlying the findings in their manuscript fully available?

Reviewer #1: Yes

Reviewer #2: Yes

5. Is the manuscript presented in an intelligible fashion and written in standard English?

Reviewer #1: Yes

Reviewer #2: Yes

6. Review Comments to the Author

Reviewer #1: I misunderstood the authors' meaning of biological validation. They used computer tools for validation and no biological experiments were performed.

Reviewer #2: In their comments and edits, the authors have successfully clarified the methods used for gene enrichment analysis. They also emphasized that PID is not a feature selection method, but rather a method for revealing higher order interactions that may uncover hidden biological insights. The biological test for PID identified genes is based on the analysis of differentially expressed genes (DEGs) and enriched pathways. Their DEG analysis uses an arbitrary absolute log fold change (LFC) cut-off of 1.5. The conclusions might be dependent on this arbitrary choice of LFC. Could the authors explore the results with no LFC cut-off and with an absolute LFC cut-off of 1?

Additionally, the identification of a small number of unique DEGs captured within the synergy cluster may not be a reliable test for biology. I urge the authors to evaluate whether the genes identified within the synergy cluster (unique and shared) show stronger associations in this group than would be expected if the groups were random (random subjects within the community, matched in size to the patients and controls in the synergy group). Observed test statistic for each gene within the synergy group can be compared to their null distributions to derive an empirical p-value for each gene.

As the paper stands, the evidence for biological insights gained by PID are subjective at best.

7. PLOS authors have the option to publish the peer review history of their article (what does this mean?). If published, this will include your full peer review and any attached files.

Reviewer #1: No

Reviewer #2: No

---

## [Author Response · Author response to Decision Letter 3]

29 Sep 2025

See file Response to Reviewers.pdf

---

## [Decision Letter · Decision Letter 3]

27 Oct 2025

Unveiling Complex Patterns: An Information-Theoretic Approach to High-Order Behaviors in Microarray Data

PONE-D-24-38840R3

Dear Dr. Monaco,

We’re pleased to inform you that your manuscript has been judged scientifically suitable for publication and will be formally accepted for publication once it meets all outstanding technical requirements.

Kind regards,

Y-h. Taguchi, Dr. Sci.

Academic Editor

PLOS ONE

Additional Editor Comments (optional):

The paper was accepted.

Reviewers' comments:

Reviewer's Responses to Questions

**Comments to the Author**

1. If the authors have adequately addressed your comments raised in a previous round of review and you feel that this manuscript is now acceptable for publication, you may indicate that here to bypass the “Comments to the Author” section, enter your conflict of interest statement in the “Confidential to Editor” section, and submit your "Accept" recommendation.

Reviewer #2: All comments have been addressed

2. Is the manuscript technically sound, and do the data support the conclusions?

Reviewer #2: Partly

3. Has the statistical analysis been performed appropriately and rigorously? 

Reviewer #2: Yes

4. Have the authors made all data underlying the findings in their manuscript fully available?

Reviewer #2: Yes

5. Is the manuscript presented in an intelligible fashion and written in standard English?

Reviewer #2: Yes

6. Review Comments to the Author

Reviewer #2: Authors have shown that PID reveals higher-order interactions relevant to disease biology. They made their statistical analysis results available, allowing readers to transparently assess the findings.

7. PLOS authors have the option to publish the peer review history of their article (what does this mean?). If published, this will include your full peer review and any attached files.

Reviewer #2: No

---

## [Editor Report · Acceptance letter]

PONE-D-24-38840R3

PLOS ONE

Dear Dr. Monaco,

I'm pleased to inform you that your manuscript has been deemed suitable for publication in PLOS ONE. Congratulations! Your manuscript is now being handed over to our production team.

Kind regards,

on behalf of

Professor Y-h. Taguchi

Academic Editor

PLOS ONE